# Exosomes Highlight Future Directions in the Treatment of Acute Kidney Injury

**DOI:** 10.3390/ijms242115568

**Published:** 2023-10-25

**Authors:** Xiaoyu Zhang, Jing Wang, Jing Zhang, Yuwei Tan, Yiming Li, Zhiyong Peng

**Affiliations:** 1Department of Critical Care Medicine, Zhongnan Hospital, Wuhan University, Wuhan 430071, China; 2019305231088@whu.edu.cn (X.Z.); wangjing9279@163.com (J.W.); jingzh46@hotmail.com (J.Z.); 2019305231095@whu.edu.cn (Y.T.); 2Clinical Research Center of Hubei Critical Care Medicine, Wuhan 430071, China; 3Department of Critical Care Medicine, Center of Critical Care Nephrology, University of Pittsburgh School of Medicine, Pittsburgh, PA 15213, USA

**Keywords:** exosomes, acute kidney injury, biomarkers, therapy

## Abstract

Acute kidney injury (AKI) is a severe health problem associated with high morbidity and mortality rates. It currently lacks specific therapeutic strategies. This review focuses on the mechanisms underlying the actions of exosomes derived from different cell sources, including red blood cells, macrophages, monocytes, mesenchymal stem cells, and renal tubular cells, in AKI. We also investigate the effects of various exosome contents (such as miRNA, lncRNA, circRNA, mRNA, and proteins) in promoting renal tubular cell regeneration and angiogenesis, regulating autophagy, suppressing inflammatory responses and oxidative stress, and preventing fibrosis to facilitate AKI repair. Moreover, we highlight the interactions between macrophages and renal tubular cells through exosomes, which contribute to the progression of AKI. Additionally, exosomes and their contents show promise as potential biomarkers for diagnosing AKI. The engineering of exosomes has improved their clinical potential by enhancing isolation and enrichment, target delivery to injured renal tissues, and incorporating small molecular modifications for clinical use. However, further research is needed to better understand the specific mechanisms underlying exosome actions, their delivery pathways to renal tubular cells, and the application of multi-omics research in studying AKI.

## 1. Introduction

Acute kidney injury (AKI) is common and is associated with high rates of mortality and disability. It has been reported that AKI occurs in approximately 20% of hospitalized patients, and in ICU patients, the incidence of AKI can even be as high as 57.3%, with a high mortality rate [1,2]. AKI can result from ischemia-reperfusion injury (IRI), trauma, sepsis, post-major surgery, and exposure to nephrotoxic substances, among others [3,4,5,6]. Due to the complex pathophysiologic mechanisms of AKI, there is currently a lack of specific drugs for its treatment, and some clinical trials have not yielded satisfactory results [5]. Exosomes (Exos), which are widely recognized as promising therapeutic choices for various diseases, hold promise in the development of new treatment strategies for AKI.

Extracellular vesicles (EVs) are a heterogeneous population of membrane-bound vesicles that have not been fully explored in terms of specific types and classifications. Currently, based on their size, biogenic processes, and molecular markers, they can be broadly categorized as follows [7]: microvesicles, exosomes, and apoptotic bodies. However, researchers have also identified some EVs that may represent novel types [8], like tumor-derived tumor bodies versus mitochondrial-derived mitotic vesicles [9]. EVs have been widely applied in regenerative medicine and nanomedicine, participating in the pathophysiology and treatment of various diseases [10,11], including the focus of our discussion, AKI [12].

Exosomes comprise an important type of EV, and they have a diameter ranging from 40 nm to 160 nm, with an average diameter of 100 nm. Exos originate from endosomes and have become an important area of research [13]. Exos were initially believed to be involved in the elimination of cellular waste. However, increasing evidence suggests that exosomes play an important role in intercellular communication. The widely recognized pathways associated with exosome uptake include receptor–ligand binding, membrane fusion, and endocytosis [14].

The contents of exosomes include proteins, lipids, and nucleic acids (such as DNA, mRNA, miRNA, and mitochondrial DNA [15]). Although the synergistic or antagonistic effects of exosome cargos are still not fully understood, numerous studies have confirmed that various cargo components of exosomes are critical to the onset and development of AKI [16].

## 2. Role of Exosomes of Different Origins in AKI

### 2.1. Role of Exosomes of MSCs in AKI

Cells of distinct types have been confirmed to secrete exosomes that may be significant in the onset and development of AKI, especially mesenchymal stem cells (MSCs). MSCs are derived from a wide range of sources, including bone marrow, adipose tissue, umbilical cord, human placenta, dental pulp, skin, blood, and urine, as well as from induced pluripotent stem cells (iPSCs) [17].

MSCs have been shown to exert prominent therapeutic effects. Results of a Phase 1 clinical trial revealed that autologous bone marrow-derived MSCs (BMMSCs) increased blood flow and the glomerular filtration rate and reduced inflammation and blood pressure in atherosclerotic renovascular disease [18]. Bone marrow-derived MSCs have been reported to repair cisplatin/glycerol-induced AKI, IRI, and unilateral ureteral obstruction (UUO)-induced renal interstitial fibrosis [19]. However, transplanted MSCs show low survival rates, restricting their therapeutic efficacy. Additionally, the paracrine effects of exosomes in regenerative medicine are of utmost importance. And a meta-analysis has shown that MSC-Exos possess therapeutic effects similar to those of MSCs in AKI [20]. Compared with stem cells, exosomes are easier to extract, store, and transport and they reflect lower immunogenicity and toxicity. They are stable in the circulation, can cross biological barriers, and can be engineered for enhanced efficacy and reduced side effects [21,22,23]. Therefore, the use of MSC-Exos as a replacement for MSCs constitutes a promising area of clinical research.

MSC-Exos have been shown to inhibit renal tubular cell apoptosis, promote tubular proliferation, alleviate inflammatory and oxidative stress, and promote vascular regeneration to repair AKI due to I/R, UUO, or cisplatin [24,25,26,27,28,29]. In addition, exosomes from BMMSCs have been found to reduce kidney fibrosis and inhibit the epithelial–mesenchymal transition in the UUO model. Newly discovered BMMSCs derived from human exfoliated deciduous teeth stem cells show high proliferative potential, self-renewal capability, and low immunogenicity [30]. Secreted exosomes can also alleviate renal injury in cisplatin-induced AKI through anti-apoptotic, anti-inflammatory, and antioxidative mechanisms [31].

Various sources of stem cells also have their unique advantages, such as ease of access, renewability, absence of ethical controversy, and low immunogenicity. Human amniotic epithelial cells (hAECs) have special advantages in clinical translation compared with BMMSCs [32]. To our knowledge, hAEC-Exos achieve protective effects in IRI, cisplatin-induced AKI, and sepsis models in mice [33,34,35]. Human placental mesenchymal stem cells (hPMSCs) are isolated from the amniotic membrane of the placenta, and the exosomes derived from hPMSCs have been shown to ameliorate renal IRI [36]. Adipose-derived mesenchymal stem cell (ADMSC) exosomes are superior to BMMSC exosomes in improving renal function [37]. The potent regenerative function of ADMSCs has attracted attention as a paracrine tool in promoting tubular regeneration in AKI. These cells have been found to alleviate inflammation and oxidative stress, promote tubular proliferation, and alleviate renal injury in sepsis and IRI models [38,39,40,41]. Furthermore, exosomes derived from ADMSCs in retrograde renal injury have been demonstrated to exert a reparative effect. In a feline model of post-renal ureteral obstruction-induced AKI, ADMSC exosome treatment effectively restored the plasma metabolome to a normal physiological steady state [42]. Human umbilical cord mesenchymal stem cell (hucMSC) exosomes regulate autophagy, reduce tubular cell apoptosis and necrosis, stabilize mitochondrial structure, and reduce oxidative stress to alleviate renal damage in cisplatin-induced AKI [43,44,45]. In IRI and UUO models, exosomes from hucMSCs were found to alter the apoptosis transcriptome in proximal tubules [45,46,47]. hucMSC exosomes have also been noted to repair renal tubular cells in an I/R-AKI model in pigs, reducing serum creatinine and blood urea nitrogen levels [48]. iPSCs possess similar characteristics to BMMSCs [49]. The secreted exosomes from iPSCs have similar renal protective properties, effectively safeguarding renal function and integrity, thereby alleviating renal IRI. Additionally, the significant induction of activated extracellular signal-related kinases 1 and 2 (ERK-1 and -2) signaling molecules can be observed [50]. Urine-derived stem cells (USCs) express the renal protective protein Klotho, and their exosomes preferentially localize to the injured kidney area. In both IRI and gentamicin-induced rat models, exosomes derived from USCs exhibit protective effects. USCs improve renal functional impairment through klotho and other similar mechanisms described above [51,52,53] (Table 1). Above all, exosomes of a variety of cellular origins have been found to have a similar regulatory effect in AKI, and in physiological and pathological states, the secretion of exosomes by cells in the body may be regulated in a similar way as hormones.

### 2.2. Role of Exosomes from Other Sources in AKI

The pathophysiologic processes of AKI do not solely rely on MSCs but also on the participation of blood cells, endothelial cells, and their exosomes. For example, investigators recently uncovered exosomes from red blood cells that showed increased levels of hemoglobin, mediating AKI after cardiopulmonary bypass in animal models, and leading to significant renal damage [60]. Macrophages, as significant inflammatory cells in AKI, have also been shown to secrete exosomes that alleviate endothelial dysfunction and reduce apoptosis and necrosis of renal tubular epithelial cells (TECs) in septic AKI through the neutrophil gelatinase-associated lipocalin-vascular cellular adhesion molecule-1 (NGAL-VCAM1) pathway [61]. Exosomes produced by pericytes may well play a critical role in maintaining the microvascular network [58]. Additionally, transplantation of endothelial progenitor cells (EPCs) significantly reduced renal tissue injury in septic rats [62], and it was shown that exosomes derived from EPCs improved the outcomes of septic mouse models by alleviating inflammation [56,59]. Exosomes from unknown sources in serum also contribute to AKI, as noted in a study in which the application of remote ischemic preconditioning had a protective effect on AKI associated with circulating exosomes [63].

Exosomes from various sources play important roles in different types of AKI, which may be related to the complex systemic effects of AKI (Figure 1). Furthermore, the mechanisms of exosomes primarily involve the following: regulating renal tubular cell proliferation and apoptosis, modulating inflammatory pathways and oxidative stress, inhibiting fibrosis progression, stabilizing endothelial cell function, and promoting angiogenesis—all of which correspond to the pathological mechanisms of AKI. This may suggest that exosomes, from a mechanistic perspective, have the potential to serve as a specific therapeutic direction for AKI.

## 3. Role of Exosome Content in AKI

To gain more detailed knowledge of the pathophysiologic mechanisms underlying AKI, researchers are conducting in-depth research on exosomes derived from different sources. Therefore, they have shifted their focus toward investigating specific small molecules within exosomes and the corresponding mechanisms by which they regulate AKI. As a result, there has been an increasing amount of research dedicated to the study of the contents of exosomes, particularly various non-coding RNAs. While there have also been some studies on proteins and lipids, their number is relatively limited at present.

### 3.1. Reparative Effects

Non-coding RNAs are abundant components in exosomes and are important in AKI [64]. MicroRNAs (miRNAs), composed of 21–25 nucleotides, induce mRNA degradation or inhibit protein translation by binding to the 3′-untranslated region (UTR) of target mRNAs. In AKI, the levels of miRNAs undergo significant changes, with the upregulation of miR-17-5p, miR-21, and others and the downregulation of miR-20a-5p and miR205, among others [64]. Additionally, depletion of Drosha in MSC-Exos inhibits their regenerative potential; and depletion of miRNAs in exosomes significantly impairs their therapeutic potential in AKI [65]. These findings highlight the important role of miRNAs in exosomes.

Different miRNAs contained in exosomes from various sources have been shown to exert reparative effects in different types of AKI by promoting tubular proliferation and repair, regulating autophagy, suppressing inflammation and oxidative stress, inhibiting fibrosis progression, and promoting angiogenesis. The role of MSC-Exos in AKI has been summarized in previous studies, and this review provides a summary and supplementation of such recent research in Table 2. As described above, exosomes have been shown to play a role in AKI, but most experiments are limited to the animal and cellular levels, only a few involve the analysis of human serum substances, and clinical trials are more rare.

### 3.2. Damaging Effects

Although numerous studies have shown that miRNAs, especially those derived from MSCs, have positive effects on AKI (including tissue repair and regeneration, restoration of cell cycle arrest, and attenuation of inflammation and fibrosis), increasing evidence suggests that miRNAs contained in exosomes are detrimental in the development of AKI, particularly in the exosomes from TECs, macrophages (Mφ), and the interplay between them.

Macrophage-derived exosomes have been shown to mediate renal glomerular endothelial dysfunction and necroptosis in sepsis-associated AKI [61,72]. They contribute to tubular cell apoptosis and inflammation in cisplatin (Cisp)-induced nephrotoxicity and IRI models [85,86]. Tubular cell-derived exosomes have also been reported to activate fibroblasts [87] through miR-150-5p, promoting renal fibrosis [88]. Proximal TEC-derived exosomes that target mitochondrial transcription factor A (TFAM) mediate Cisp-induced TEC injury [89]. Furthermore, communication between TECs and macrophages through exosomes has been demonstrated to strongly facilitate the progression of AKI. TEC-derived exosomes carrying microRNA-374b-5p regulate macrophage polarization, promoting M1 macrophage activation and thus worsening AKI [90]. Another study revealed that exosomes derived from lipopolysaccharide (LPS)- or hypoxia-inducible factor-1α-stimulated TECs enhanced the activation of M1 macrophages and mediated renal injury in an AKI mouse model [91,92]. Moreover, activated macrophage-derived exosomes are internalized by tubular cells, leading to increased tubular injury [86]; and this may result in a vicious cycle that develops into a cascade of AKI.

In addition to TECs and macrophages, neutrophils have also been found to promote AKI progression through exosomes. In vitro and in vivo experiments have confirmed that neutrophil-derived exosomes containing miR-30d-5p induce M1 macrophage polarization by upregulating NLRP3 inflammasome expression via the nuclear factor κB (NF-κB) signaling pathway, targeting suppressors of cytokine signaling 1, and sirtuin 1 [93] (Table 3).

Due to the complex bidirectional interactions between exosomes and their cargo, the clinical timing of exosome administration becomes crucial. Modulating the content and structure, or modifying the bioactive substances in a precise manner at the appropriate time, can achieve more accurate therapeutic effects while minimizing potential side effects in AKI. However, all of these interventions should be based on a more comprehensive understanding of the physiologic effects of each cargo and a thorough knowledge of the pathological processes of the disease, which requires further investigation. CircRNAs act as regulatory factors of miRNAs and can affect the expression of target genes by binding to miRNAs. They also function as important components of exosomes to regulate various pathological processes. For example, highly stable circ-FANCA is prominently expressed in sepsis-associated AKI patients and LPS-stimulated HK2 cells. Knocking down circ-FANCA delivered through exosomes alleviated HK2 cell damage in LPS-induced septic AKI by targeting the miR-93-5p/OXSR1 axis [96]. Circ_0001818 has also been found to activate the expression of TXNIP, leading to LPS-induced HK2 cell damage [97].

Long non-coding RNAs (lncRNAs) can regulate the stability of miRNAs and also play a crucial role as contents of exosomes. lncRNAs are also involved in AKI, including tubular programmed cell death and inflammation. The expression of lncRNA TUG1 is downregulated in LPS-induced and IRI-induced AKI. At the same time, the upregulation of the TUG1 gene shows a protective effect on IRI-induced kidney injury. Additionally, lncRNA TUG1 from USCs exosomes regulates the stability of ACSL4 mRNA by interacting with SRSF1, thereby regulating ferroptosis and reducing renal IRI [79].

In addition to their functions through non-coding RNAs, exosomes can also exert information exchange by directly transferring mRNA. For example, it has been reported that exosomes isolated from bovine serum albumin-pretreated renal tubular epithelial cells enhanced inflammation and macrophage migration by transferring CCL2 mRNA, thereby enhancing proteinuria-induced kidney injury [81].

Proteins, another major component of exosomes, have been found to wield functions such as adsorption, promoting intercellular information, and receptor–ligand binding in the process of AKI. Previous studies had shown that exosomes derived from hucMSCs carrying 14-3-3ζ had significant therapeutic effects on tissue regeneration in cisplatin-induced AKI [82]. It has also been reported that ASC-EVs target damaged kidneys in a CD44-dependent manner [98]. A single-center prospective cohort study showed that CD26 of urinary exosomes was correlated with the recovery of AKI in the intensive care unit [99]. MSC-derived exosomes overexpressing polyamine oxidase may also be associated with regulating macrophage polarization (the M1 to M2 transition) and mediating the repair process in IRI/AKI [100].

As noted above, most of the current studies focus on miRNAs in AKI, while other non-coding RNAs, mRNAs, and even less-studied components such as mitochondria, proteins, and lipids are largely overlooked. These molecules may also play important roles in AKI, and, therefore, further research is needed in the future.

## 4. Exosomes and Their Contents Act as Biomarkers for AKI

The current definition of AKI is still limited to parameters such as serum creatinine and urine output measurements, which provide standardized criteria to some extent, but these markers lack sensitivity. Although several new biomarkers have been proposed, their distribution over time was influenced by various factors, and they also had limitations. Exosomes are widely present in bodily fluids, and the contents of exosomes remain relatively stable under extreme conditions, including high RNase activity, low or high pH, long-term room temperature storage, and multiple freeze–thaw cycles [101,102]. Therefore, using exosomes and their contents as potential biomarkers has attracted immense interest.

In recent years, miRNAs derived from exosomes have been explored as biomarkers for AKI. One study showed that miRNA analysis of urinary exosomes can be used to assess the progression of AKI. During injury, miR-16, miR-24, and miR-200c were elevated in urine, while during early recovery, miRs (miR-9a, miR-141, miR-200a, miR-200c, and miR-429) were upregulated together [103]. In the analysis of early biomarkers for LPS-induced AKI in rats, it was found that miR-181a-5p and miR-23b-3p were expressed at higher levels in the LPS-treated group than in the control group [104]. Urinary exosomal microRNA-21 has also been proposed as a biomarker for AKI associated with scrub typhus [105].

In addition to miRNAs, proteins contained in exosomes have been identified as potential biomarkers. One research study showed that urinary exosomes from vancomycin-induced AKI patients exhibited significantly upregulated expression of inflammatory proteins following nephrotoxic injury [106]. An analysis in 2022 revealed urinary exosomal NHE3 in rats with various types of AKI and in 12 patients, suggesting its potential as a diagnostic biomarker for AKI [107]. Furthermore, the combination of uEV-AQP2 and -AQP1 had been suggested to facilitate the estimation of cisplatin-induced decompensated renal injury [108].

Considering that exosomes exist in body fluids, show stability, and are capable of predicting cellular status and reflecting disease development based on their quantity and content, exosomes and their contents have been widely explored as novel non-invasive biomarkers for various diseases. However, routine monitoring methods may be limited by either no detection or low detection efficiency, thus driving the development of innovative detection methods for exosome enrichment and high-efficiency testing to achieve accurate early diagnosis.

In recent years, a rapid isolation system called EXODUS has been developed for exosome detection, enabling fast and efficient separation and detection of exosomes [109]. Another method is surface plasmon-coupled electrochemiluminescence analysis based on plasma nanostructures; this method consists of a polarized electrochemiluminescence sensor for exosomal miRNA detection and has shown significantly improved detection efficiency [110]. Single-exosome counting has also achieved the detection of only 10 enzyme-labeled exosomes per microliter of serum [111]. Furthermore, breakthroughs have been made in the detection limits of individual exosomes using hollow-core, anti-resonant optical fibers. Additionally, microfluidic-based liquid biopsies have been used for exosome separation and detection and have achieved high sensitivity [112]. The continuous development of emerging technologies has improved the detection rate of exosomes, lowered the detection threshold, and achieved high specificity and accuracy, thus promoting the clinical application of exosomes as biomarkers and facilitating early diagnosis of various diseases.

## 5. Engineering Exosomes

Although there is considerable research indicating the promising application of exosomes and their contents for the treatment of AKI, there are currently no standardized protocols for the large-scale production, efficient targeting, standardized separation and purification methods, and storage conditions for exosomes. This signifies that the quality of exosomes cannot be controlled. And problems associated with the large-scale production of exosomes like production costs and efficiency needed to be solved. Therefore, it is necessary to develop engineered exosomes or utilize modifications on exosomes to achieve their separation and enrichment, and to enhance their targeting ability with respect to injured tissues.

### 5.1. Isolation and Enrichment

The most commonly used method for exosome isolation is differential ultracentrifugation, which is applied by approximately 80% of researchers worldwide [113]. However, due to the similarities in size characteristics among different types of exosomes, this method can only achieve a crude separation and lacks additional purification. Other methods such as ExtraPEG, immune-affinity chromatography (IAC), and size-exclusion chromatography (SEC) all have advantages and limitations [114,115]. Some studies have suggested that sequentially combining multiple isolation methods can increase exosomal purity, but issues of cost and quantity still require further discussion [116].

Exosome isolation techniques are constantly evolving and show numerous innovations. Tangential flow filtration (TFF) is more suitable for the large-scale production of higher quantities and activity than ultracentrifugation isolation [117]. The ExoSCRT™ technology based on TFF can extract over 4 L of ASC-CM within 4 h, enabling the reproducible production of exosomes with comparable impurity levels that comply with WHO standards [118]. A 3D culture system based on hollow-fiber bioreactors has additionally been implemented to produce MSC-Exos, showing a 19.4-fold improvement in MSC-Exo yield [119].

Exosome enrichment can also be achieved through methods such as transferring target peptide expression plasmids into cells to generate exosomes with target ligands, directing the loading of drugs via electroporation, and creating mixed exosomes through freeze–thaw cycles. Squeezing drug-loaded cells through a series of filters with reduced pore sizes produces biomimetic exosomes [120]. While these novel technologies are still under development, their application in AKI remains limited. Thus, new technologies for exosome isolation and enrichment and their applications to other diseases should also be specified and optimized for AKI.

### 5.2. Targeted Therapy

The improvement of the targeting ability of exosomes to an injured site is currently being conducted to enhance the efficacy of exosomes in AKI treatment. For example, P-selectin has been identified as a biomarker for endothelial cell ischemic AKI. In one study, the authors applied a novel binding method to covalently link evPBP (CDAEWVDVS) with DMPEPEG5000-maleimide (DMPE-PEG-MAL), resulting in the synthesis of DMPEPEG-PBP (DPP). DPP can then be inserted into the membrane of Exos to selectively target the injured kidney. Molecular imaging was thus adopted to quantitatively assess the expression of P-selectin in the kidney, demonstrating a protective effect [121]. Another approach involved the use of peptides or antibodies that specifically bind to target molecules. For instance, the LTH peptide (which binds to Kim1) was conjugated to red blood cell-derived EVs (REVs) using a bifunctional cross-linker that enabled its targeting to the kidney [122]. Furthermore, neutrophil membrane-engineered NEX significantly promoted the targeted enrichment of exosomes in damaged renal tissue, leading to improvements in AKI [123]. Researchers have also discovered that the functionalization of MSC-derived EVs (MSC-EVs) using monocyte membranes through membrane fusion significantly enhanced their targeted capability in damaged tissues [124]. However, these applications are currently focused on cardiac diseases [125], and their usage in AKI has yet to be explored to any great degree. Due to the similarity in pathological damage between AKI and heart disease (i.e., inflammation or mediation by IRI), it is speculated that the therapeutic effects may also be comparable. In addition, various hybridization methods from tumor-therapy modalities could offer valuable insights for targeted exosome therapy in AKI. These methods include the hybridization of EVs with synthetic liposomes, coupling between post-modified diazonium compounds and antibodies that are modified with dibenzocyclooctyne (DBCO) [126], coupling with diacyllipid-DNA aptamers (sgc8) [127], anchoring EVs to superparamagnetic nanoparticles [128], and the induction of electrostatic interactions to facilitate fusion between cationic lipids and exosomes [129].

In summary, the aforementioned methods in tumor or heart disease provide insights into the precise targeted treatment and monitoring of AKI using Exos.

### 5.3. Exosome Modifications

Exosome modifications can be achieved through gene editing and chemical drug modifications. Exos can serve as carriers to deliver nucleic acids for gene editing, or they can be genetically engineered to incorporate gene sequences that encode specific membrane proteins. However, this technology is limited to currently known gene sequences.

Pre-treatment with small-molecule drugs is a common method to enhance therapeutic effects. Compared with genetic manipulation, small-molecule drugs reflect significant advantages: their effects can be fine-tuned by altering their working concentration, duration, and composition. Researchers have found that resveratrol enhanced the restorative effects of HucMSC exosomes in cisplatin-induced AKI [130]. Similarly, 3,3′-dimethylthiazolyl diphenyl tetrazolium bromide stimulated the self-secretion of Wnt11 in MSC-derived exosomes and promoted wound healing [131]. Supramolecular nanofibers containing arginine-glycine-aspartate (RGD) peptides improved the therapeutic effects of EVs in kidney repair [132], and platelet-rich plasma induced YAP nuclear expression to promote BMMSC proliferation, activating the AKT/Rab27 pathways to facilitate the paracrine secretion of MSC-derived exosomes for glycerol-induced AKI repair [133]. Additionally, hypoxia-induced activation of HIF-1α was also found to increase the loading of miR-21 in exosomes, thereby providing a therapeutic effect against septic/IRI-induced AKI [134].

Enhancing the functionality of exosomes through small-molecule pre-treatment is widely recognized. Although numerous molecules have been evaluated, further exploration is needed to determine the optimal timing and underlying mechanisms of action of small-molecule interventions.

## 6. Future Directions

### 6.1. Accelerating Clinical Translation

We have already illustrated the meaningful role of exosomes in AKI, but there are still many challenges to achieving clinical translation. Currently, most experiments are limited to animals or just cellular levels, only a few involve analysis of human serum substances, and clinical trials are even more rare. What is more, one important reason for fewer clinical trials is the shortage of engineering exosomes, lower purity, separation efficiency. So, we would like to find new markers not inferior to the used Kim-1 or P-selectin in AKI and then design specific binding ligands with packaged engineering to implement precise and efficient targeted treatment and monitoring of AKI. On this basis, we would accelerate clinical translation to achieve early diagnosis and treatment of AKI, but at the same time, we cannot ignore potential regulatory hurdles or ethical concerns; although, exosome research involves less ethics than stem cells. It still involves a range of ethical issues, such as informed consent of donors; privacy and personal information confidentiality of donors; exosomes of animal origin; and the participation of ethical review committees and regulatory bodies in exosome research is crucial.

When exosomes are clinically applied, then as drugs, the potential differences in exosome treatment responses of different patients are inevitable. These individual differences can be influenced by a variety of factors, including genetic, epigenetic, and other factors. Some studies have identified genetic variants associated with exosome treatment response, such as mutations in certain genes that may lead to increased resistance or sensitivity of cells to treatment. In addition, epigenetic differences may also lead to different responses to exosome therapy. Furthermore, other factors such as chronic disease, drug use, and immune status may also affect a patient’s response to exosome therapy. And these factors may alter the sensitivity or resistance of cells to treatment. Patients may develop an immune response to treatment, resulting in exosomes being cleared or losing function. More research is needed to understand how these factors affect the effectiveness of exosome treatment.

### 6.2. Elucidating Pathways of Reaching the Renal Tubules

Tubular injury is a significant aspect of AKI. Many studies have shown that exosomes exert their effects by entering renal tubular epithelial cells; however, further exploration is required to understand how exosomes reach damaged renal tubules. While there is evidence that exosomes can cross the blood–brain barrier, there is limited research on assessing whether exosomes can pass through the glomerular filtration membrane in the kidneys. Concerning the mechanisms by which exosomes reach the renal tubular cells, two hypotheses are proposed. (1) Exosomes enter the primary urine through the glomerular filtration membrane and are then reabsorbed by the renal tubules; and (2) exosomes cannot pass through the glomerular filtration membrane but instead interact with the tubules through a network that involves afferent arterioles, efferent arterioles, and peritubular capillaries. It is also possible that both pathways co-exist simultaneously. The process by which systemically administered exosomes enter the vicinity of the renal tubules thus requires further investigation. We postulate that the development of stable in vivo imaging techniques will facilitate the elucidation of this complex and intricate process.

### 6.3. Exosomes and Multi-Omics Studies

Current research on exosomes is limited, with analyses focused on miRNAs by matching them with potential target mRNAs to elucidate the function of Exo-associated miRNAs. However, many pathways in the body are intertwined and regulated by multiple factors, making it difficult to fully understand the complexity of diseases using this approach. Research on Exos is still evolving, and multi-omics analysis has become a major area of current work. The combined application of exosomes and multi-omics analysis may well bring us closer to the essence of disease in general and provide a comprehensive understanding of cellular changes. However, as the combined application of Exos and multi-omics analyses in kidney diseases has not yet been explored, we recommend that greater attention be given to this area.

## 7. Conclusions

This review provides an overview of the mechanisms underlying exosome activity, particularly those derived from blood cells and mesenchymal cells, in AKI. We herein explored the reparative effects of various exosome cargo components in AKI and highlighted the interactions between renal tubular cells and macrophages through exosomes that promote the progression of AKI. Furthermore, this review comprised the clinical potential of exosomes and their contents as biomarkers for the early diagnosis and staging of AKI. We also discussed exosome engineering research in AKI, including isolation and enrichment, targeted therapy, and molecular modifications, as well as research progress in tumor and heart disease. Finally, this review comprised ideas for overcoming the limitations of exosome research in AKI. However, elucidation of the molecular underpinnings of how exosomes reach the renal tubular cells and the combined application of exosomes and multi-omics studies await further investigations in the future.

## Figures and Tables

**Figure 1 ijms-24-15568-f001:**
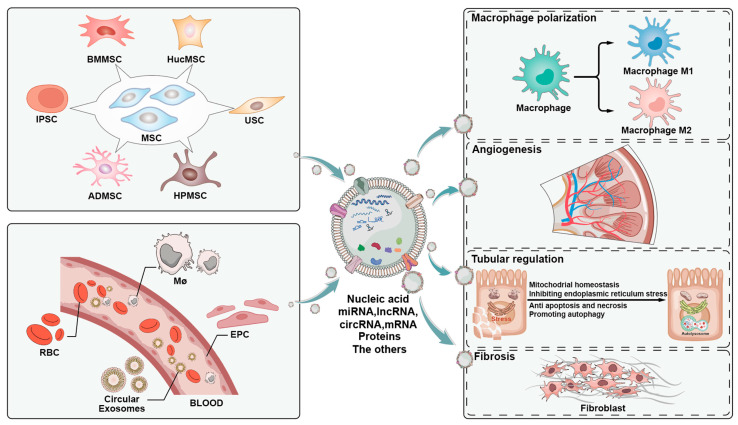
Exosomes from various sources including red blood cells, macrophages, monocytes, and mesenchymal stem cells in promoting renal tubular cell regeneration and angiogenesis, regulating autophagy, suppressing inflammatory responses and oxidative stress, and preventing fibrosis to facilitate AKI repair through their contents (such as miRNA, lncRNA, circRNA, mRNA, and proteins).

**Table 1 ijms-24-15568-t001:** Mechanisms of exosomes of different origins in AKI.

Exo Source	Model	Mechanism	Pathways/Markers	Reference
BMMSC	IRI Rats Mice HK-2 (H/R)	Anti-apoptotic Reduces oxidative stress Anti-inflammatory Promote angiogenesis Inhibits endoplasmic reticulum stress	IL-6, TNF-α, NF-κB IFN-γ ↓ caspase-9, cleaved caspase-3, Bax, and Bcl-2 ↓ BIP miR-149/let-7c/Faslg	[24,54,55]
BMMSC	UUO Rats	Inhibits fibrosis Epithelial mesenchymal transformation Anti-inflammatory Anti-apoptotic	STAT3 ↓ MMP-9P Colla1,α-SMA ↓ PCNA ↓	[19]
BMMSC	Cisp Mice	Anti-apoptotic Reduces DNA damage	Inhibiting p53	[27]
SCAP	Cisp NRK-52E	Anti-apoptotic Reduces oxidative stress Anti-inflammatory	NF-κβ, IL-1β, p53 ↓ Bcl-2 ↑ Bax, CASP8, CASP9, and CASP3 ↓	[31]
hAEC	IRI mice HK-2 (H/R)	Anti-apoptotic Anti-inflammatory Promote angiogenesis	extracellular matrix growth factor cytokine production, immunomodulation	[33]
hAEC	Sepsis LPS-treated HUVECs CLP mice	Anti-inflammatory Maintained endothelial cell adhesion junction Mitigating endothelial dysfunction	phosphor-p65, p65, VCAM-1, ZO-1	[35]
hAEC	Cisp mice	Anti-inflammatory	TNF-α/MAPK and caspase signaling ↓	[34]
ADMSC	IRI	Anti-apoptotic Anti-inflammatory Reduces oxidative stress, DNA damage Promote angiogenesis Inhibits fibrosis Relieves tubular damage	TNF-α/NF-κB/IL-1β/MIF/PAI-1/Cox-2 ↓ NOX-1/NOX-2/oxidized protein ↓ Bax/caspase-3/PARP ↓ Smad3/TGF-β ↓	[38]
ADMSC	Sepsis CLP mice LPS HK-2	Anti-apoptotic Anti-inflammatory Promote angiogenesis Autophagy	SIRT1 TLR9	[37,39,41]
ADMSC	Postrenal AKI Cats	Metabolome	carnitine, melibiose, D-Glucosamine, cytidine, dihydroorotic acid, stachyose ↑	[42]
USCs	Plasma from AKI patients IRI rats IRI mice HK-2 (H/R)	Anti-inflammatory Inhibits fibrosis Reduces oxidative stress	TRAF6 IRAK1 NF-κB p65 IL10 and TGFβ1 ↑ IFN-γ and IL-1β ↓	[51,52,53]
IPSC	IRI HK-2 (H/R)	Reduce renal damage	ERK 1/2 signaling	[50]
HuMSC	Cisp NRK-52E cells	Promotes TEC proliferation Anti-apoptotic Anti-inflammatory Reduces oxidative stress Promotes mitochondrial fusion Autophagy	fibronectin, α-SMA, GSDMD, caspase-1, IL-1β and NLRP3 ↓	[56]
HuMSC	UUO rats	Inhibits fibrosis	CK1δ/β-TRCP inhibited YAP activity	[47]
HuMSC	IRI rats	Anti-inflammatory Inhibits fibrosis Reduces oxidative stress		[45,46,57]
Macrophages	Sepsis CLP mice	Relieves endothelial cell dysfunction Reduces tubular cell apoptosis, pyroptosis	HMGB, VCAM1 ↓	[56]
Pericytes	IRI mice	Reduces tubular cell damage Maintain capillary stability	Cadherin APN	[58]
EPC	Sepsis CLP rats	Anti-inflammatory	RUNX1 ↓	[59]

Abbreviations: BMMSC, bone marrow mesenchymal stem cell; ADMSC, adipose-derived mesenchymal stem cell; SCAP, apical papillary stem cell; USCs, urine-derived mesenchymal stem cells; IPSC, induced pluripotent stem cell; hAEC, human amniotic mesenchymal stem cell; EPC, endothelial progenitor cell; VCAM-1, vascular cell adhesion molecule-1; VCAM1, vascular cell adhesion molecule 1; APN, Adiponectin; ↓, Decreased expression; ↑, increased expression.

**Table 2 ijms-24-15568-t002:** Reparative role of exosomal contents in AKI.

Content	Exo Source	Model	Mechanism	Pathways	Reference
miR-199a-3p	BMMSC	IRI mice HK-2 (H/R)	Anti-apoptotic Reduces oxidative stress	Sema3A ↓—ERK, AKT ↑	[24]
miR-let-7b-5p	BMMSC	Cisp mice	Anti-apoptotic Reduces DNA damage	p53 ↓—DNA damage and apoptosis pathway activity ↓	[27]
miR-1184	BMMSC	HK-2 by cisp	Anti-apoptotic Anti-inflammatory Break the G1 block	Targeting FOXO4, p27 Kip1 and CDK2	[66]
miR-146b	HucMSC	LPS HK-2 CLP mice	Anti-inflammatory	IRAK1 ↓ NF-κB ↓	[67]
miR-486-5p	HucMSC	IRI mice	Promotes proliferation angiogenesis	PTEN ↓—Akt phosphorylation	[45]
miR-486-5p	HucMSC	IRI mice	Anti-apoptotic Anti-inflammatory	protein kinase B ↑ TNF pathway ↓ phosphatase and tensin homolog↓	[68]
miR-125b-5p	HucMSC	IRI mice HK-2 (H/R)	Break the G2/M block Anti-apoptotic	P53 ↓—CDK1 and Cyclin B1 ↑ modulation of Bcl-2 and Bax accumulated in proximal tubules by virtue of the VLA-4 and LFA-1	[46]
miR-874-3p	HucMSC	UUO mice HK-2 by cisp	Reduce necrosis Promotes mitochondrial fusion	RIPK1 PGAM5 ↓—dephosphorylation of the S637 site of the Drp1 gene	[69]
miR-342-5p	ADMSC	Sepsis patients with sepsis-associated AKI CLP mice HK-2 by LPS	Enhanced autophagy Anti-inflammatory Reduced BUN and SCr levels	TLR9 ↓—autophagy ↑ BUN, SCr, ↓	[41]
miR-216a-5p	USC	IRI rats HK-2 (H/R)	Promote proliferation Angiogenesis	PTEN ↓—Akt phosphorylation	[70]
miR-146a-5p	USC	IRI rats HK-2 (H/R)	Anti-inflammatory Anti-apoptotic	Target 3′UTR of IRAK1—NF-κB ↓	[52]
miR-30a-5p	USCp	Cisp mice HK-2 by cisp	Anti-apoptotic Anti-inflammatory	MAPK8 ↓	[71]
miR-93-5p	Macrophage (M2 > M1)	CLP mice TCMK-1 cells by LPS	Renal epithelial cell pyroptosis Anti-inflammatory Anti-apoptotic	TXNIP—pyroptosis in renal epithelial cell	[72]
miR-590-3p	TEC	AKI patients after cardiac surgery HK-2 (H/R)	Autophagy	miR-590-3p was highly enriched in the plasma exosomes of young AKI patients after cardiac surgery Beclin-1 and LC3II ↑	[53]
miR-20a-5p	TEC H	HK-2 (H/R) IRI mice	Promotes TEC proliferation, Promotes mitochondrial fusion, Attenuating necrosis	macrophages infiltration ↓	[73]
miR-21-5p	ECFC	CLP mice	Anti-inflammatory Promotes proliferation angiogenesis	RUNX1 ↓	[60]
miR-486-5p	ECFC	IRI mice	Promotes proliferation angiogenesis	PTEN ↓—Akt phosphorylation	[45]
miR-93-5p	EPC	HK-2 by LPS CLP mice	Anti-apoptotic Anti-inflammatory	Regulating KDM6BH/3K27me3/TNF-α axis	[74]
miR-124	Unknown	CIS mice	Anti-inflammatory	MCP-1 ↑	[75]
miR500a3p	Unknown	HK-2 by cisp	Reduce necrosis	RIPK3 and MLKL ↓	[76,77]
miR-150-5p	Unknown	HK-2 by LPS LPS mice	Anti-apoptotic Anti-inflammatory and oxidative stress	MEKK3/JNK pathway	[78]
lncRNA TUG1	USC	HK-2 (H/R) IRI mice	Reduce Ferroptosis	ACSL4 ↑—TUG1’s repression ↓ miR-494-3p—E-cadherin and TUG1	[79] [80]
CCR2	BMMSC	IRI mice	Inhibits macrophage recruitment activation	Unknown	[81]
14-3-3z	HucMSC	NRK-52E by cisp Cisp Rats	Autophagy	ATG16L↑	[82]
CD26	TEC	IRI mice	Anti-inflammatory Promotes TEC proliferation	CXCR4, SDF1 ↓	[83]
Klotho	USC	AKI rats	Anti-apoptotic Anti-inflammatory Promotes proliferation angiogenesis	fibrosis, monocyte infiltration ↓ SOD1 ↑	[84]

Abbreviations: BMMSC, bone marrow mesenchymal stem cell; ADMSC, adipose-derived mesenchymal stem cell; EPC, endothelial progenitor cell; Sema3A, semaphorin 3A; AKT, activated protein kinase B; ERK, extracellular-signal-regulated kinase; FOXO4, forkhead box O4; IRAK1, interleukin (IL)-1 receptor-associated kinase; PTEN, Phosphatase and tensin homolog; PGAM5, phosphoglycerate mutase 5; RIPK1, Receptor-interacting protein kinase 1; MAPK8, mitogen-activated protein kinase 8; LC3II, microtubule-associated protein 1 light chain 3 beta; RUNX1, runt-related transcription factor 1; SDF1, stromal derived factor-1; SOD1, superoxide dismutase 1; TEC H, hypoxic tubular epithelial cell; ECFC, endothelial colony-forming cell; CIS, Candida infection with sepsis; USCp, premature USC; ↓, Decreased expression; ↑, increased expression.

**Table 3 ijms-24-15568-t003:** Damaging role of exosomal contents in AKI.

Content	Exo Source	Model	Mechanism	Pathways	Reference
miR-195a-5p	Macrophage	Cisp mice	Promote TEC apoptosis and damage TEC mitochondria	unknown	[85]
miR-155	Macrophage	IRI/cisp mice HK-2 (cisp) NRK-52E (H/R)	Promote TEC damage, promote inflammation limiting the telomeric dysfunction and the genomic DNA damage	TRF1, CDK12 ↓ TCF4/Wnt/β-Catenin	[86,94,95]
miR-19b-3p	LPS-TEC	Sepsis LPS mice	Promotes inflammation, macrophage 1 polarization	NF-κB/SOCS-1	[91]
miR-150-5p	TEC	IRI UIRI rats NRK-52E (H/R)	Activates fibroblasts and promotes fibrosis	suppressor of cytokine signaling 1 to activate fibroblast ↓	[88]
miR374b-5p	TEC	IRI mice	Promote Macrophage 1 polarization	Transferring miR-374b-5p	[90]
miR-709	PTC	Cisp mice human AKI kidney	Promote TEC apoptosis and mitochondria damage	TFAM ↓ miR-709 in PTCs of patients ↑	[89]
miR-23a	TEC H	UUO/IRI mice and TEC	Promotes inflammation Macrophage activation	suppression of the ubiquitin editor A20	[92]
miR-30d-5p	Neutrophils mice	Sepsis	Promotes inflammation, macrophage 1 polarization Promotes pyroptosis of macrophages	NLRP3—NF-κB ↑ SOCS-1 and SIRT1 ↓	[93]
Circ-FANCA	Sepsis patient LPS-HK2	Sepsis LPS HK-2	Promotes inflammation, apoptosis Inhibit proliferation (G0/G1 arrest)	sponging OXSR1	[96]
Circ0001818	Unknown (Serum)	LPS-HK2	Promote LPS-HK2 damage	miR-136-5p—TXNIP	[97]

Abbreviations: TEC H, hypoxic tubular epithelial cell; PTCs, proximal tubular cells; TFAM, mitochondrial transcription factor A; LPS, Lipopolysaccharides, ↓, Decreased expression; ↑, increased expression.

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
