# Peer review of "Exosomes Highlight Future Directions in the Treatment of Acute Kidney Injury"

_ijms, 2023, doi:10.3390/ijms242115568_

Round 1

Reviewer 1 Report

This review article examines the complex role of exosomes from different cell sources in AKI pathology. It highlights their reparative mechanisms as well as damaging effects mediated through exosomal content transfer between cells. The article also discusses their potential as diagnostic biomarkers and engineered therapeutic agents, while identifying key areas for future research to advance exosome-based therapies for AKI.

  1. The sources and purity of extracted exosomes are variable between different studies. Standardized isolation and characterization methods should be used.
  2. The specific pathways of how exosomes reach injured renal tubular cells are still unclear. Advanced in vivo imaging techniques could help elucidate the transport mechanisms.
  3. Most of the research has focused on miRNA contents of exosomes. Other components like proteins, lipids, lncRNAs should also be explored.
  4. The interactions between different exosome contents that lead to synergistic or antagonistic effects are not well understood.
  5. The timing of exosome administration can be crucial due to their complex effects. Optimal therapeutic windows need to be defined.
  6. Large-scale production and storage methods for clinical-grade exosomes remain challenging. More efficient bioreactor systems could help.
  7. Specificity of exosomes to target injured tissues is low. Engineered exosomes with targeting moieties need to be developed.
  8. Toxicity and immunogenicity of modified exosomes must be thoroughly evaluated before clinical use.
  9. A systems biology approach combining exosome analysis with genomics, proteomics, metabolomics data could provide greater insights into disease mechanisms and exosome effects.

Author Response

Thank you for your review report sincerely. The concrete response could be viewed at the attachment. Please see the attachment.

Reviewer 2 Report

While the review provides a comprehensive overview of the role of exosomes in treating acute kidney injury (AKI), there are several areas where it can be further improved.

Most of the paper revolves around pre-clinical and cellular models. It would be beneficial to discuss any available clinical trials or patient data to validate the therapeutic potential of exosomes in human AKI patients.

The paper doesn't delve deep into specific protocols for therapeutic application. For instance, what would be the recommended dose of exosomes, mode of administration, and potential side effects or contraindications?

There is no discussion comparing the efficacy of exosomes derived from different cell sources. Do they all equally benefit AKI, or are some sources more effective than others?

There's no discussion on the scalability, production costs, or challenges associated with the large-scale production of exosomes for therapeutic purposes.

The paper doesn't address potential regulatory hurdles or ethical concerns associated with using exosomes, especially those derived from stem cells or other human/animal sources.

The review doesn't address the potential variability in response to exosome therapy among different patients. Are there genetic, epigenetic, or other factors that might influence the therapeutic outcomes?

While the paper touches upon the stability of exosomes, it would be beneficial to delve deeper into the best practices for their storage, transport, and shelf-life, especially if they are to be considered for widespread therapeutic use.

Any potential therapy can have adverse reactions. A section discussing potential risks or side effects associated with exosome therapy would be crucial.

Could there be mechanisms by which AKI (or cells involved) become resistant to exosome therapy? This is an area worth exploring.

The review could benefit from placing the potential of exosomes in AKI treatment within the broader context of other emerging therapies for AKI. How do they compare in terms of efficacy, safety, and feasibility?

The overall quality of english is good.

Author Response

(The authors gave the same response as above.)

Reviewer 3 Report

I have had the opportunity to thoroughly review the manuscript with the identifier ijms-2649478, which offers a comprehensive look into the burgeoning field of exosomes as a therapeutic avenue for acute kidney injury (AKI). The article is well-structured, meticulously covering the various subtypes of exosomes, their key biological attributes, and their interactions with renal cells. Additionally, the manuscript provides a nuanced exploration of exosome modification strategies, including gene editing techniques and small-molecule interventions. A salient feature of the article is its discussion on the application of exosomes in targeted drug delivery, which holds considerable promise based on existing pre-clinical evidence. While optimistic, the review responsibly identifies hurdles to clinical translation, such as the current reliance on animal or cellular-level studies and challenges in exosome purification and targeting. The article concludes with a forward-looking perspective, calling for accelerating clinical trials, clarifying the mechanisms that facilitate exosome delivery to renal tubules, and incorporating multi-omics approaches for a more holistic understanding of AKI. In summary, the article underscores the potential utility of exosomes in AKI treatment and emphasizes the necessity for ongoing, in-depth research to capitalize on this potential fully.

1. Does Exos mean exosome? at line 50, 71, 74,75,84,138,143,255,268,285,288,332,334

2. Delete of Line 244 of 4.1. Isolation and enrichment, if this is a typographical error

3. Line 300, "IRIAKI," please show the full name before the abbreviation.

4. A double period appears at the end of line 394.

5. In Figure 1, Is Mo means macrophages or monocytes? If it is macrophages, please modify the symbol to MÏ•.

Author Response

(The authors gave the same response as above.)

Round 2

Reviewer 2 Report

Authors clarified the comments raised by the reviewer

English quality is good